# Daily versus stat vitamin D supplementation during pregnancy; A prospective cohort study

Nida Bokharee[1], Yusra Habib Khan[2]*, Tayyiba Wasim[3], Tauqeer Hussain Mallhi[2]*, Nasser Hadal Alotaibi[2], Muhammad Shahid Iqbal[4], Kanwal Rehman[5], Abdulaziz Ibrahim Alzarea[2], Aisha Khokhar[1]

1 Institute of Pharmacy, Lahore College for Women University, Lahore, Pakistan, 2 Department of Clinical Pharmacy, College of Pharmacy, Jouf University, Sakaka, Al-Jouf, Kingdom of Saudi Arabia, 3 Department of Gynaecology, Services Institute of Medical Sciences, Services Hospital, Lahore, Pakistan, 4 Department of Clinical Pharmacy, College of Pharmacy, Prince Sattam bin Abdulaziz University, Al-kharj, Saudi Arabia, 5 Department of Pharmacy, University of Agriculture, Faisalabad, Pakistan

* yusrahabib@ymail.com (YHK); tauqeer.hussain.mallhi@hotmail.com (THM)

**Data Availability Statement:** All relevant data are within the manuscript.

**Funding:** The authors received no specific funding for this work.

## Abstract

### Background

Despite favorable climatic conditions, vitamin D deficiency (VDD) is widespread in Pakistan. Current study was aimed to evaluate the prevalence of VDD in Pakistani pregnant women and effectiveness of various regimen of Vitamin D supplementation.

### Methodology

This hospital-based prospective cohort study included pregnant women at 12th to 24th weeks of gestation attending Gynae clinic from October 2018 to April 2019. Patients were classified into control and treatment groups (Groups: $G_1$, $G_2$ and $G_3$) according to the dose of vitamin D supplementation. Patients received various regimens of vitamin D including 2000 IU/day ($G_1$), 5000 IU/day ($G_2$) and stat 200000 IU ($G_3$). The levels of vitamin D were measured before and after supplementation. The effectiveness of dosages were compared between and within the groups. Moreover, factors associated with vitamin D sufficiency and insufficiency were ascertained using appropriate statistical methods.

### Results

Among 281 pregnant women (mean age: 28.22 ± 4.61 years), VDD was prevalent in 47.3% cases. Vitamin D supplementation caused significant rise in the levels 25(OH)D in treatment groups, while there was no significant difference in control group. The highest mean increment in vitamin D (23.14 ± 11.18 ng/ml) was observed with dose 5000 IU/day followed by doses 200000 IU stat (21.06 ± 13.73 ng/ml) and 2000 IU/day (10.24 ± 5.65 ng/ml). Vitamin D toxicity was observed in one patient who received 200000 IU stat of vitamin D. The frequency of VDD following the supplementation was 5.7%. Education status, duration of sun exposure and use of sunblock was substantially associated with vitamin D sufficiency in the current study.

**Competing interests:** The authors have declared that no competing interests exist.

## Conclusion

Our findings underscore the high proportion of VDD among pregnant women in Pakistan. Maternal vitamin D supplementation substantially improved the levels of 25(OH)D. Of three used regimens, the dose of 5000 IU/day is considered safe and equally effective as of 200000 IU stat. Since pregnancy is a time of tremendous growth and physiological changes for mother and her developing fetus with lifelong implications for the child, gestational vitamin D supplementation should be considered to ensure the optimal vitamin D accrual in pregnant women. This study generates the hypothesis that vitamin D supplementation at a dose of 5000 IU/day during pregnancy is superior to the other regimens. However, well-controlled randomized trials are needed to confirm these findings.

## Introduction

Vitamin D deficiency (VDD) is widespread around the globe and associates with negative maternal and neonatal health outcomes. Despite the geographical region with a warm climate, VDD is widely prevalent in Pakistan. The National survey conducted in 2011, concluded that 68.9% pregnant women were vitamin D deficient [1]. Majority of the women in Pakistan practice Hijab (veil) due to religious or cultural reasons, and spend most of their time indoor and thus have predilection to suffer from VDD [2]. Though the definition of VDD is quite debated but plasma levels of 25-hydroxyvitamin D (25(OH)D) < 20 ng/ml is widely accepted in the literature [2–4].

Vitamin D can be naturally obtained from direct sun-exposure to skin (Ultra Violet-B radiations). Sunlight converts 7-dehydrocholesterol to pre-vitamin $D_3$ in the skin, which is further metabolized to vitamin $D_3$ and then to 25(OH)D in the liver. Renal conversion of 25(OH)D to 1, 25-dihydroxyvitamin $D_3$ (1, 25-OH2D$_3$) maintains calcium hemostasis [2, 5]. Sun exposure of face and forearms at mid-day for about 20–30 minutes produces around 2000 IU equivalent of vitamin D in light-fair skinned population. However, duration of sun exposure is 2 to 10 times for dark skinned population to produce the equivalent amount of vitamin D [6]. Dietary sources of Vitamin D include oily fish, egg yolk, milk, juices, yogurts, cereals, soy, mushrooms, margarine and cod liver oil. VDD can lead to musculoskeletal manifestations such as osteomalacia in adults, rickets in children, metabolic disorders (secondary hyperparathyroidism), obstetric complications such as pre-eclampsia (PET), gestational diabetes mellitus (GDM) and gestational hypertension (GHT), increased probability of cesarean section, pre-term delivery, and decreased bone mineral density (BMD) [7].

Vitamin D supplementation is not routinely recommended during antenatal care, as there is not enough evidence to support its benefits during pregnancy [2, 8–11]. The Recommended Dietary Allowance (RDA) in pregnancy is 600 IU (15 mcg) and 400 to 600 IU according to Institute of Medicine (IOM) [12]. However, revision of guidelines during gestation and lactation were suggested by several investigators that supplementation must be evidence-based and in accordance with the clinical relevance [13]. For modestly dressed pregnant female, 1000 IU (25mg) per day is recommended due to inadequate sun-exposure [14]. The dose of 2000 IU/day is also considered safe but inadequate in most of the studies and therefore the dose of 4000 IU was preferred [5, 15–20]. Daily tolerable upper intake limit according Institute of medicine (IOM) is 4000 IU and according to The Endocrine Society Clinical Practice Guidelines by Holick et al. (2011) is 10,000 IU and no evidence of toxicity was associated at these doses [2, 4, 21, 22].

Increased prevalence of VDD globally and its associated health related intricacies have raised a major concern and hence needs to be addressed, especially in developing countries such as Pakistan. However, there is unavailability of regional data on high dose supplementation during gestation. Current study was aimed to ascertain the prevalence of VDD during pregnancy, effectiveness of various regimens of Vitamin D supplementation (200000 IU single dose, daily high dose of 2000 IU and 5000 IU) and proportion of the study population attaining sufficient vitamin D levels following treatment.

## Methodology

### Ethical approval

This study was approved by the Mid City Hospital's Ethical Review Board (Reference: MCH/ EXC/CEO-01). Informed written consent was obtained and purpose of the study was explained to the participants. All the patient's identities were anonymised before analysis.

### Study design and location

This hospital-based prospective cohort study was conducted in the Outpatient Department (OPD) of the Mid City Hospital (MCH), a multi-disciplinary hospital known due its specialty in Gynaecology which serves hundreds of patients daily.

### Study population

All the pregnant women at 12-24th weeks of gestation attending OPD of MCH during October 2018 to April 2019 were consecutively recruited into the study. Gestational age was calculated in weeks on the basis of Last Menstrual Period (LMP). Fetal ultrasound was also done to ensure the gestational age or any other anomalies. Pregnant women with renal disease, chronic Liver disease (CLD), or those using anti-tubercular or anti-epileptic drugs during last three months were excluded as they can affect the study outcomes.

### Treatment groups

Since vitamin D screening is routinely performed for patients registered in the hospital, the baseline levels of vitamin D were available for all patients. Pregnant women were classified by the researcher into four different groups according to the dose of Vitamin D prescribed. The choice of vitamin D supplementation and dose was at the discretion of the individual treating physician. Patients in which vitamin D supplementation was initiated were classified into three groups (Supplementation Groups (G) i.e. $G_1$, $G_2$ and $G_3$) according to the dose they received. Patients in $G_1$ received 2000 IU/day, $G_2$ received 5000 IU/day and $G_3$ received 200,000 IU single stat dose. Patients who were not prescribed any vitamin D dose were classified as control group (CG). Patients in CG received conventional antenatal management. In our hospital, patients are encouraged to report any adverse event to the antenatal clinic or directly to the pharmacist and side effects for overdose were monitored in all patients receiving supplementation during the study period. A follow up was scheduled two months after initiation of dose, in compliance with the current antenatal care follow up visit. The process of study flow is described in Fig 1.

### Outcome measure

Serum 25(OH)D level was used as the measuring outcome to assess the vitamin D status at baseline and follow-up to compare the effectiveness of prescribed supplementation.

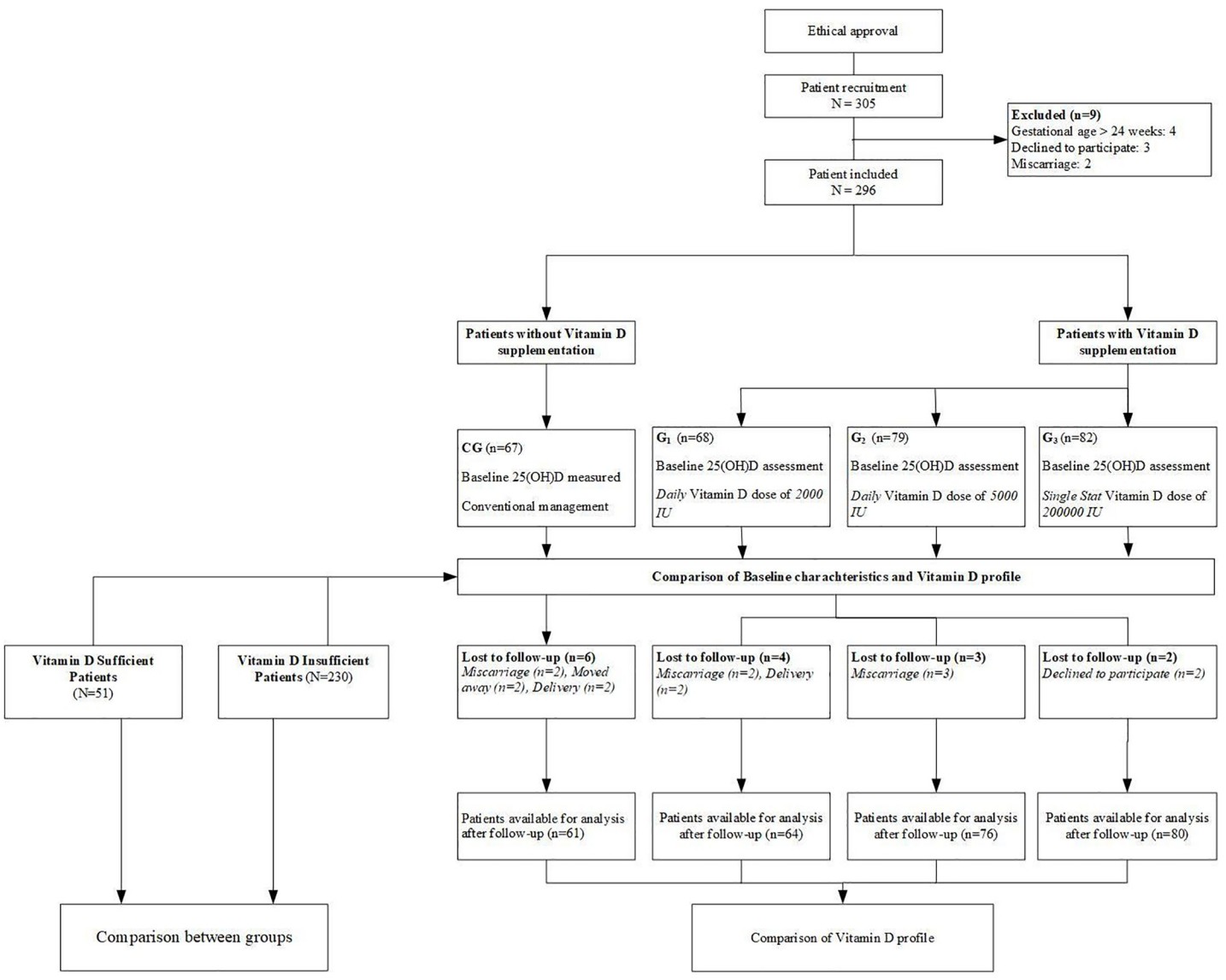

**Fig 1. Study flow diagram.**

## Safety measures

Vitamin D toxicity was defined as circulating level of 25(OH)D >100 ng/ml. Vitamin $D_3$ supplementation was stopped in case of toxicity. Hypercalcemia was measured using serum calcium level and routinely Ultrasonography (USG) was conducted for the high-risk patients to observe the kidney stones formation. Patients were monitored followed by the supplementation for the rest of the study period (2 months).

## Biochemical analysis

Vitamin D status was evaluated by measuring serum 25(OH)D level. Maternal blood sample was collected, centrifuged and stored at -80°C followed by Chemiluminescence or CLIA (Chemiluminescence Immunoassay Analyzer) using state of the art Maglumi® 600 fully automated system. Maternal blood (5 milliliters) was collected at baseline and again at the follow-

up. CLIA is a quantitative method which measures total 25(OH)D and other hydroxylated vitamin D metabolites in serum sample. CLIA is two-incubation assay in which antibody-antigen complex is formed; 25(OH)D is dissociated from the binding protein followed by its binding to 25(OH)D antibody. The chemiluminescent reaction was as relative light units which are inversely proportional to the 25(OH)D in the sample. Serum Vitamin D concentration was measured in Nano grams per milliliter (ng/ml). The cut-off reference points used to define vitamin D status in this study were < 20 ng/ml as deficiency, 20 to < 30 ng/ml as insufficiency, 30–100 ng/ml as sufficiency and > 100 ng/ml as toxicity [2–4, 23–31]. Maternal serum calcium levels were assessed using spectrometry method at the follow-up to rule out any manifestation of vitamin D intoxication.

## Data collection

Data Collection was devised to gather the information regarding demographics, gestation, parity of recruits, medical history, medication history, Clinical features indicating osteomalacia (muscle weakness, bone pain, tenderness, or fractures). Patient's demographics were recorded directly from patients and their medical records. Patient compliance to the regimen was assessed by self-reporting.

## Statistical analysis

An IBM SPSS version 25 was used to perform all statistical analysis. The data was recorded as the Mean ± standard deviation (SD) for the continuous variables and as frequencies with percentages (proportion) for the categorical variables. The comparisons between more than two treatment groups for normally distributed data was done using one way ANOVA or Kruskal Wallis test, where appropriate. The comparison (univariate) between two categorical variables and dichotomous data was carried out using $\chi^2$ (chi square) or Fisher Exact test, where appropriate. Comparison of vitamin D levels between baseline and follow-up within each treatment groups was made using paired t-test. Comparison of patients with vitamin D sufficiency and insufficiency was conducted by chi-square test for categorical variables. Chi-square was used to check the association between the educational status and self-medication. A logistic regression model was performed to determine the factors independently associated with vitamin D insufficiency. Odds ratio and 95% confidence interval were also calculated. One-way ANCOVA was performed to compare different interventions and to control the effect of confounders. A two-tailed p value of 0.05 was considered significant.

## Results

### Characteristics of study participants

Out of 305 patients, 296 were recruited into the study. Of these, 9 patients were excluded from the analysis due to their refusal to participate in the study (n = 3), miscarriage (n = 2) and gestational age > 24 weeks (n = 4) (Fig 1). A total of 281 patients completed the study and were available for the analysis. Of these, 61 patients were in control group, 64 in $G_1$, 76 in $G_2$ and 80 patients were in $G_3$ group.

The mean maternal and gestational age of study participants was 28.2 ± 4.6 years and 18.2 ± 4.2 week, respectively. The demographics were equally distributed between the control and treatment groups. Approximately half of the study participants (49.5%) were over-weight, 172 (61.2%) were graduates, 201 (71.5%) were housewives and 280 (99.6%) were Asian. Thirty two (11.4%) pregnant women reported self-medication of analgesics (n = 25), folic acid (n = 4) and multivitamins (n = 3). Demographics and clinical features were equally distributed

between the study groups. The vitamin D rich food consumption was also shown to be equally distributed between the treatment groups. The patients were recruited in three different season with decreased UV index (UVI) and were equally distributed across the treatment groups; (Autumn = 22nd September– 21st December, Winter = 22nd December– 20th March, Spring = 21st March - 21st June) (Table 1). The mean 25(OH)D level was lowest in winters (20.46 ± 10.53 ng/ml), with concentration recovering in spring (20.60 ± 9.70 ng/ml) and highest in the current study in autumn (23 ± 8.31 ng/ml). Moreover, there was no statistically significant difference between the mean 25(OH)D level between the groups (p = 0.149).

## Impact of supplementation on 25(OH)D levels

Table 2 demonstrates that the baseline levels of 25(OH)D were equally distributed between control and treatment groups (p = 0.245). During follow-up, the levels of 25(OH)D were significantly improved in treatment groups, with the highest mean serum 25(OH)D achieved in $G_2$ group. Sub-group analysis showed that there was no statistically significant ($P = 0.686$) difference of vitamin D levels between 5000 IU/day (43.92 ± 16.95 ng/ml) and 200,000 IU stat (41.50 ± 15.33 ng/ml) regimens. The highest proportion of patients (78.8%) achieved sufficient levels of 25(OH)D were in G3 group. Our results showed that supplementation improved the proportion of patients with vitamin D sufficiency from 18.1% to 65.8%. Only one patient attained serum 25(OH)D > 100 ng/ml. However, serum biochemical indices were within the normal range and USG showed no stones in the kidneys. Levene's test and normality checks were carried out and assumptions met.

All the confounding variables i.e. sun-exposure, seasonal variation, use of sunblock, body area covered and baseline VD level were adjusted using General linear model to assess the effect of intervention. One-way ANCOVA was conducted to compare the effectiveness of different interventions (CG, G1, G2, G3) on patient's VD level. There is no significant relationship between the covariate and the dependent variable, after controlling for the independent variable (treatment group) and adjusting covariates i.e. baseline VD level, seasonal variation, sun-exposure, body area covered and use of sunblock. There was a strong relationship between the baseline and follow-up VD level, as indicated by a partial eta squared value of 0.375 (Table 3).

Supplementation with different vitamin D doses had a variable effect on circulating vitamin D. Current study demonstrated significant increment in serum vitamin D level in treatment groups following supplementation (Table 4). However, the increase in the levels of vitamin D in control group was insignificant ($p = 0.061$). The highest mean increment (23.14 ± 11.18 ng/ml) was observed with dose 5000 IU/d followed by dose 200000 IU stat (21.06 ± 13.73 ng/ml) ($p < 0.001$). The Vitamin D increment was statistically different between the control group and treatment [p-value: CG and $G_1$: 0.001; CG and $G_2$: < 0.001; CG and $G_3$: 0.001]. Moreover, the VD increment in G1—$G_2$ and G1—$G_3$ were also statistically different (p-value < 0.001). However, VD increment was statistically insignificant between $G_2$ and $G_3$ (p = 0.579).

## Risk factors of vitamin D insufficiency among study participants

Table 5 indicates the factors associated with vitamin D sufficiency and insufficiency. The patients who had vitamin D sufficiency (30–100 ng/ml) (n = 51) and those with insufficiency (n = 230) were compared with each other. There was a significant difference between the two groups for their educational status. Patients with lower education level, sun exposure for less than 30 minutes or no sun-exposure were associated with vitamin D deficiency. In the sufficiency group 74.5% were graduate and 27.5% had a daily sun-exposure for more than 1 hour.

**Table 1.** Demographic characteristics of the treatment groups recorded at baseline.

| Characteristics | Total (N = 281) | Control (n = 61) | G$_1$ (n = 64) | G$_2$ (n = 76) | G$_3$ (n = 80) | P value |
|---|---|---|---|---|---|---|
| **Maternal age** (years ± SD) | 28.22 ± 4.61 | 27.61 ± 4.18 | 28.41 ± 5.09 | 28.71 ± 4.80 | 28.09 ± 4.36 | 0.552 |
| **Maternal age range**, n (%) | | | | | | 0.320 |
| 18–27 years | 128 (45.6%) | 33 (54.1%) | 27 (42.4%) | 33 (43.4%) | 37 (46.3%) | |
| 28–37 years | 138 (49.1%) | 28 (45.9%) | 34 (53.1%) | 36 (47.4%) | 37 (46.3%) | |
| 38–47 years | 15 (5.3%) | 0 | 3 (4.7%) | 6 (7.9%) | 6 (7.5%) | |
| **Education; n (%)** | | | | | | 0.428 |
| Un-educated | 4 (1.4%) | 0 | 2 (3.1%) | 2 (2.6%) | 0 | |
| Primary | 2 (0.7%) | 0 | 1 (1.6%) | 1 (1.3%) | 0 | |
| Matric | 7 (2.5%) | 3 (4.9%) | 2 (3.1%) | 1 (1.3%) | 1 (1.3%) | |
| Intermediate | 38 (13.5%) | 10 (16.4%) | 10 (15.6%) | 12 (15.8%) | 6 (7.5%) | |
| Graduate | 172 (61.2%) | 34 (55.7%) | 39 (60.9%) | 51 (67.1%) | 48 (60.0%) | |
| Post-Graduate | 58 (20.6%) | 14 (23.0%) | 10 (15.6%) | 9 (11.8%) | 25 (31.3%) | |
| **Occupation; n (%)** | | | | | | 0.052 |
| Un-employed | 201 (71.5%) | 44 (72.1%) | 40 (62.5%) | 64 (84.2%) | 53 (66.3%) | |
| Student | 14 (5.0%) | 5 (8.2%) | 3 (4.7%) | 4 (5.3%) | 2 (2.5%) | |
| Professional | 64 (22.8%) | 12 (19.7%) | 19 (29.7%) | 8 (10.5%) | 25 (31.3%) | |
| Self-employed | 2 (0.7%) | 0 | 2 (3.1%) | 0 | 0 | |
| **Gestational age** (weeks) | 18.21 ± 4.17 | 17.77 ± 4.05 | 18.30 ± 4.20 | 17.47 ± 4.23 | 19.16 ± 4.04 | 0.064 |
| **BMI** | 24.82 ± 4.40 | 25.12 ± 3.78 | 24.53 ± 4.75 | 24.63 ± 4.31 | 24.99 ± 4.69 | 0.844 |
| **BMI categories** | | | | | | 0.655 |
| Underweight | 14 (5.0%) | 1 (1.6%) | 3 (4.7%) | 3 (3.9%) | 7 (8.8%) | |
| Normal | 77 (27.4%) | 21 (34.4%) | 19 (29.7%) | 20 (26.3%) | 17 (21.3%) | |
| Over-weight | 139 (49.5%) | 29 (47.5%) | 32 (50.0%) | 38 (50.0%) | 40 (50.0%) | |
| Obese | 51 (18.1%) | 10 (16.4%) | 10 (15.6%) | 15 (19.7%) | 16 (20.0%) | |
| Gravidity (Median) | 2 | 2 | 2 | 2 | 2 | |
| **Total Number of pregnancies** | | | | | | 0.676 |
| PG/None | 130 (46.3%) | 25 (41%) | 30 (46.9%) | 39 (51.3%) | 36 (45%) | |
| MG/More | 151 (53.7%) | 36 (59%) | 34 (53.1%) | 37 (48.7%) | 44 (55%) | |
| Parity (Range) | 0–4 | 0–4 | 0–4 | 0–3 | 0–4 | |
| **Mode of last delivery**; n (%) | | | | | | 0.206 |
| Normal | 72 (25.6%) | 16 (26.2%) | 20 (31.3%) | 19 (25.0%) | 17 (21.3%) | |
| C-section | 62 (22.1%) | 13 (21.3%) | 18 (28.1%) | 12 (15.8%) | 19 (23.8%) | |
| **Milk Consumption** (per day) | | | | | | 0.078 |
| None | 111 (39.5%) | 28 (45.9%) | 26 (40.6%) | 36 (47.4%) | 21 (26.3%) | |
| Once | 128 (45.6%) | 27 (44.3%) | 28 (43.8%) | 32 (42.1%) | 41 (51.2%) | |
| Twice | 42 (14.9%) | 6 (9.8%) | 10 (15.6%) | 8 (10.5%) | 18 (22.5%) | |
| **Fish Consumption** (per day) | | | | | | 0.530 |
| None | 273 (97.2%) | 58 (95.1%) | 63 (98.4%) | 75 (98.7%) | 77 (96.3%) | |
| Once | 8 (2.8%) | 3 (4.9%) | 1 (1.6%) | 1 (1.3%) | 3 (3.8%) | |
| **Egg Consumption** (per day) | | | | | | 0.053 |
| None | 122 (43.4%) | 25 (41.0%) | 30 (46.9%) | 43 (56.6%) | 24 (30%) | |
| Once | 137 (48.8%) | 30 (49.2%) | 29 (45.3%) | 30 (39.5%) | 48 (60%) | |
| Twice | 22 (7.8%) | 6 (9.8%) | 5 (7.8%) | 3 (3.9%) | 8 (10%) | |
| **Recruitment Season** | | | | | | 0.072 |
| Autumn | 112 (39.9%) | 24 (39.3%) | 24 (37.5%) | 25 (32.9%) | 39 (48.8%) | |
| Winter | 104 (37.0%) | 19 (31.1%) | 30 (46.9%) | 27 (35.5%) | 28 (35.0%) | |

(*Continued*)

**Table 1.** (Continued)

| Characteristics | Total (N = 281) | Control (n = 61) | G$_1$ (n = 64) | G$_2$ (n = 76) | G$_3$ (n = 80) | P value |
|---|---|---|---|---|---|---|
| Spring | 65 (23.1%) | 18 (29.5%) | 10 (15.6%) | 24 (31.6%) | 13 (16.3%) | |

P values are calculated between control and treatment groups using $\chi^2$ or Fisher Exact Test (categorical variables) and One-way ANOVA or Kruskal-Wallis Test (continuous variable), where appropriate

Gravidity: Total pregnancies, regardless of outcome; Parity: Number of births after 24 weeks, live or still birth

Abbreviations: G: Treatment Group; PG (Primigravida)–First time pregnancy; MG (Multigravida)—Multiple pregnancies, regardless of outcome; BMI (Body Mass Index): < 18.5 kg/ m$^2$ as under-weight, 18.5–22.9 kg/m$^2$ as Normal weight, 23.0–24.9 kg/m$^2$ as over-weight and $\geq$ 25.0 kg/m$^2$ as obese (Asian cut-off values)

To identify possible risk factors of vitamin D insufficiency among pregnant women, a series of logistic regression analysis was performed for clinically relevant and statistically tested variables (Table 6). Out of five tested variables, average daily sun-exposure (OR: 14.8, $p$ = 0.009) and use of sunblock (OR: 4.4, p = 0.045) were two factors with a higher likelihood of vitamin D insufficiency. Patients with average daily sun exposure less than 15 minutes and those using sun block while going outside presented a higher risk of vitamin D insufficiency in this study. Seasonal variation was adjusted as covariate, there was no significant result of season on the baseline vitamin D levels.

## Discussion

To the best of our knowledge, this is the first study to explore the impact of various vitamin D antenatal supplementation regimens among pregnant women in Pakistan. The key findings of the present study demonstrated the high prevalence of VDD during gestation. Women

**Table 2. Vitamin D status between the treatment groups.**

| Measure | Total (N = 281) | Control (n = 61) | G$_1$ 2000 IU/day; (n = 64) | G$_2$ 5000 IU/day; (n = 76) | G$_3$ 200000 IU stat; (n = 80) | P Value |
|---|---|---|---|---|---|---|
| | | | **Baseline** | | | |
| 25(OH)D (mean ± SD); ng/ml | 21.51 ± 10.49 | 23.82 ± 9.41 | 21.51 ± 8.87 | 20.77 ± 2.46 | 20.43 ± 10.34 | 0.245 |
| **Vitamin D status** | | | | | | **0.005** |
| Deficiency < 20 | 133 (47.3%) | 16 (26.2%) | 30 (46.9%) | 44 (57.9%) | 43 (53.8%) | |
| Insufficiency 20 to < 29.9 | 97 (34.5%) | 29 (47.5%) | 24 (37.5%) | 17 (22.4%) | 27 (33.8%) | |
| Sufficiency 30–100 | 51 (18.1%) | 16 (26.2%) | 10 (15.6%) | 15 (19.7%) | 10 (12.5%) | |
| Toxicity > 100 | 0 | 0 | 0 | 0 | 0 | |
| | | | **Follow-up** | | | |
| 25(OH)D (mean ± SD); ng/ml | 36.85 ± 15.16 | 27.29 ± 10.82 | 31.75 ± 8.41 | 43.92 ± 16.95 | 41.50 ± 15.33 | **<0.001** |
| **Vitamin D status** | | | | | | **0.016** |
| Deficiency < 20 | 16 (5.7%) | 12 (19.7%) | 2 (3.1%) | 2 (2.6%) | 0 | |
| Insufficiency 20 to < 30 | 79 (28.1%) | 24 (39.3%) | 24 (37.5%) | 15 (19.7%) | 16 (20.0%) | |
| Sufficiency 30–100 | 185 (65.8%) | 25 (41.0%) | 38 (59.4%) | 59 (77.6. %) | 63 (78.8%) | |
| Toxicity > 100 | 1 (0.4%) | 0 | 0 | 0 | 1 (1.3%) | |

Sufficient vitamin D concentration = 30–100 ng/ml

P values are calculated between the groups.

G$_1$ group with 2000 IU/day dose

G$_2$ group with 5000 IU/day dose

G$_3$ group with 200000 IU stat dose

Control group with no treatment

P values are calculated between control and treatment groups using one-way ANOVA

**Table 3. Relationship of confounding factors with respect to treatment groups.**

| COVARIATES | Estimated Marginal mean | p-value | Partial eta squared |
|---|---|---|---|
| **Seasonal variation** | | 0.288 | 0.012 |
| Autumn | 37.32 ± 1.45 | | |
| Winter | 37.00 ± 1.49 | | |
| Spring | 34.00 ± 1.76 | | |
| **Average sun-exposure** | | 0.240 | 0.006 |
| < 15 min | 37.21 ± 1.26 | | |
| > 15 min | 34.95 ± 1.37 | | |
| Use of sunblock | 36.88 ± 1.93 | 0.607 | 0.001 |
| **Body area covered** | | 0.758 | 0.000 |
| Fully covered | 36.56 ± 1.74 | | |
| Partially covered | 35.94 ± 1.00 | | |
| **Baseline VD level** | | 0.245 | 0.375 |
| Control | 24.73 ± 1.87 | | |
| G1 | 31.40 ± 2.16 | | |
| G2 | 43.07 ± 1.77 | | |
| G3 | 42.23 ± 1.48 | | |

P values are calculated using, General Linear Model (one-way ANCOVA) to adjust the confounding variables

classified to $G_1$, $G_2$ and $G_3$ groups when compared to those receiving no treatment experienced improved vitamin D status during the follow-up of 2 months. The high proportion of VDD in study population can be attributed to the various cultural, social, demographic and socioeconomic factors.

Despite adequate sunlight, high prevalence of VDD is reported in Pakistan. The reported prevalence of VDD as of 47.3%, insufficiency of 34.5% and sufficiency of only 18.1% is similar to the results of previously conducted studies in Pakistan [32–34]. Speculating from the results of the current study, VDD is probably much higher at national level than that reported in this study. The antenatal vitamin D supplementation is mainstay of therapy and widely recommended. On the other hand, comparatively higher prevalence (89% to 99.5%) of VDD was reported in some studies [33, 35]. Methodological variations among available studies led to the great disparity in the incidence as well as the epidemiology of VDD, making it difficult or even impossible to compare findings across the studies. Such varying prevalence might be attributable to several factors including different inclusion criteria with variable gestational age and BMI, variation in population with respect to financial status (our study site receives financially stable patients), different laboratory techniques for the estimation of serum 25(OH)D,

**Table 4. Mean increment in the vitamin D level after supplementation (within the group analysis).**

| Group | Mean 25(OH)D at baseline (ng/ml) [a] | Mean 25(OH)D at follow-up (ng/ml) [a] | Mean increment in 25(OH)D level [a] (ng/ml)* | t value | p value |
|---|---|---|---|---|---|
| **CG** | 23.82 ± 9.41 | 27.29 ± 10.82 | 3.47 ± 6.41 | 4.23 | **0.061** |
| **G₁** | 21.51 ± 8.87 | 31.75 ± 8.41 | 10.24 ± 5.65 | 14.49 | **< 0.001** [b] |
| **G₂** | 20.77 ± 12.46 | 43.92 ± 16.95 | 23.14 ± 11.18 | 18.05 | **< 0.001** [b] |
| **G₃** | 20.43 ± 10.34 | 41.50 ± 15.33 | 21.06 ± 13.73 | 13.72 | **< 0.001** [b] |

* Mean increment (from baseline to follow-up) in vitamin D serum concentration is measured as ng/ml

[a] Data is tabulated as Mean ± SD

[b] Paired t-test significant value of < 0.05

**Table 5. Comparison between patients having sufficient and insufficient status of vitamin D at baseline.**

| Characteristics | Sufficient (n = 51) | Insufficient (n = 230) | p value |
|---|---|---|---|
| **Maternal age range;** n (%) | | | 0.308 |
| 18–27 years | 21 (41.2%) | 107 (46.5%) | |
| 28–37 years | 29 (56.8%) | 109 (47.4%) | |
| 38–47 years | 1 (2.0%) | 14 (6.1%) | |
| **BMI** | | | 0.153 |
| Under-weight | 0 | 14 (6.1%) | |
| Normal | 12 (23.5%) | 65 (28.2%) | |
| Over-weight | 31 (60.8%) | 108 (47.0%) | |
| Obese | 8 (15.7%) | 43 (18.7%) | |
| **Mode of last delivery;** n (%) | | | 0.671 |
| Normal | 13 (25.5%) | 59 (25.7%) | |
| C-section | 9 (17.6%) | 53 (23.0%) | |
| **Total Number of pregnancies** | | | 0.421 |
| PG/None | 21 (41.2%) | 109 (47.4%) | |
| MG/More | 30 (58.8%) | 121 (52.6%) | |
| **Education;** n (%) | | | **0.004** |
| Un-educated | 0 | 4 (1.7%) | |
| Primary | 0 | 2 (0.9%) | |
| Matric | 4 (7.8%) | 3 (1.3%) | |
| Intermediate | 6 (11.8%) | 32 (13.9%) | |
| Graduate | 38 (74.5%) | 134 (58.3%) | |
| Post-Graduate | 3 (5.9%) | 55 (23.9%) | |
| **Occupation;** n (%) | | | 0.506 |
| Un-employed | 40 (78.4%) | 161 (70.0%) | |
| Student | 3 (5.9%) | 11 (4.8%) | |
| Professional | 8 (15.7%) | 56 (24.3%) | |
| Self-employed | 0 | 2 (0.9%) | |
| **Average daily sun exposure** | | | **< 0.001** |
| None | 5 (9.8%) | 81 (35.2%) | |
| < 15 min | 1 (2.0%) | 53 (23.0%) | |
| 15–30 min | 9 (17.6%) | 79 (34.3%) | |
| 31–60 min | 22 (43.1%) | 15 (6.52%) | |
| > 1 hour | 14 (27.5%) | 2 (0.9%) | |
| **Body area covered** | | | 0.440 |
| Partially covered | 44 (86.3%) | 188 (81.7%) | |
| Fully covered | 7 (13.7%) | 42 (18.3%) | |
| **Sunblock use** | | | **0.027** |
| Yes | 2 (3.9%) | 36 (15.7%) | |
| No | 49 (96.1%) | 194 (84.3%) | |
| **Use of Fish** | 0 | 8 (3.5%) | 0.358 |
| **Use of Egg** | 27 (52.9%) | 132 (57.4%) | 0.562* |
| **Use of Milk** | 28 (54.9%) | 142 (61.7%) | 0.366 |
| **Recruitment season** | | | 0.198 |
| Autumn | 26 (51.0%) | 86 (37.4%) | |
| Winter | 15 (29.4%) | 89 (38.7%) | |

(*Continued*)

**Table 5.** (Continued)

| Characteristics | Sufficient (n = 51) | Insufficient (n = 230) | p value |
|---|---|---|---|
| Spring | 10 (19.6%) | 55 (23.9%) | |

*Abbreviations*: PG (Primigravida)–First time pregnancy; MG (Multigravida)—Multiple pregnancies, regardless of outcome

Chi-square test or Fisher Exact test was used to assess the association of variables

*Fisher exact test, while all other p values are from Chi-square test

different cut-off references for VDD and sufficiency, and inconsistent VDD definitions. There is an on-going debate on utility of criteria for vitamin D status [36, 37]. However, majority of the studies suggest the level of < 20 ng/ml for 25(OH)D as a cut-off value for VDD [23–26].

A recent systematic review concluded that VDD is highly prevalent and supplementation proved to be an effective intervention during gestation in Pakistan [38]. Sun-exposure and vitamin D rich diet alone cannot maintain adequate levels in pregnant women. Food fortification and creating awareness through public health programs will be of paramount importance to curb the growing burden of VDD in Pakistan. However, vitamin D supplementation is required in addition to diet and sun-exposure to achieve optimal concentration. It is evident from the previous investigations that vitamin D supplementation during pregnancy can improve both maternal and neonatal status for vitamin D [38]. The findings of these studies corroborate with our results. Socio-religious restrictions or limited outdoor activity results in decreased sun-exposure and in majority of cases even low dose of 600 IU is not prescribed. Present study used high doses of vitamin D including 2000 IU/day, 5000 IU/day and 200000 IU stat. It is evident from the previous studies that high dose of 200,000 IU is effective and considered safe [39–43].

Our findings indicate that vitamin D supplementation significantly increases the levels of serum 25(OH)D during pregnancy, particularly if the supplementation regimen was daily versus stat. However, this response was highly heterogeneous in different studies [15, 44–46]. Increment in 25(OH)D in $G_2$ group receiving daily supplement was higher as compared to group receiving single dose ($G_3$). It is important to note that in daily supplemented groups ($G_1$

**Table 6. Risk factors for vitamin D insufficiency by regression model.**

| | Univariate analysis | | | Multivariate analysis | | |
|---|---|---|---|---|---|---|
| Variables | P value | OR | 95% CI for OR | P value | OR | 95% CI for OR |
| Education Level | 0.765 | 0.9 | 0.41–1.9 | - | - | - |
| Average daily Sun-exposure | **0.008** | 14.9 | 2.1–110.9 | **0.009** | 14.8 | 2.0–109.7 |
| Milk Consumption | 0.367 | 1.3 | 0.7–2.5 | - | - | - |
| Egg Use | 0.562 | 1.1 | 0.7–2.2 | - | - | - |
| Use of sunblock | **0.042** | 4.6 | 1.1–19.5 | **0.047** | 4.4 | 1.1–19.3 |

p-values with > 0.250 were excluded from the Multivariate analysis

Odds Ratio (OR) and Confidence Interval (CI) have been rounded off

Codes for logistic regression

Education level—0: intermediate level or less, 1: graduation level or above

Average daily sun exposure—1: 0: More than 15 minutes per day, Less than 15 minutes per day

Milk consumption—0: No, 1: Yes

Egg use—0: No, 1: Yes

Use of sunblock while going outside—0: No, 1: Yes

and $G_2$), only high dose of 5000 IU ($G_2$) showed higher serum 25(OH)D level in the current study.

Similar to the earlier studies, the beneficial effects of the dose 2000 IU/day are evident from the present study [15, 17, 18, 20, 47, 48]. However, the results and conclusions are heterogeneous, several studies concluded that 2000 IU/day dose do not achieve sufficiency in majority of the patients [17]. Despite the rise in serum 25(OH)D levels with the use of 2000 IU group, majority of the patients remained insufficient in their vitamin D status and hence higher dose should be preferred and recommended. Moreover, where low dose of vitamin D is recommended, patients should be encouraged and counselled to increase their daily sun-exposure up to 1 hour.

In the present study, the dose of 5000 IU/day was used to achieve the optimal vitamin D status. Yap *et al.*, (2014) conducted a study on high and low daily doses of vitamin D and showed significantly higher plasma 25(OH)D levels achieved with 5000 IU/day dose. Authors concluded that supplementation with 5,000 IU/day vitamin $D_3$ during pregnancy can safely and effectively elevate the serum 25(OH)D concentrations into the desired target range in 90% of the women [49]. In another similar study, 97% of women attained vitamin D concertation as of 80 nmol/l (32 ng/ml) at the time of delivery with dose 5000 IU/day [50]. These results are consistent with the findings of the present study and dose of 5000 IU/day was concluded as safe and effective to achieve the optimal concentration of 25(OH)D.

Existing data indicate that the high dose of 200,000 IU stat is effective and safe to achieve desired vitamin D status [44, 51]. These results are in line with our findings where sufficient serum 25(OH)D status was achieved in maximum number of patients. Our results in corroboration with other studies suggest that dose of 200,000 IU is effective and carries the advantage of compliance. Moreover, patients prescribed with high dose of vitamin D should be monitored for their serum 25(OH)D levels. It is pertinent to mention that one safety measure was taken and further supplementation was stopped as per safety protocol in the $G_3$ group. Serum vitamin D concentration was 108 ng/ml in this patient. Further investigations revealed that patients was taking drug at multiple times along with other multivitamins. Fortunately, the serum biochemical indices were within the normal range and USG findings indicates no stones in either kidneys. These findings suggest the periodic monitoring of patients receiving high dose. These patients must be educated to avoid concurrent use of other multivitamins, excessive sun-exposure, use of sunblock and to monitor any unwanted effects. The baseline 25(OH)D levels should be estimated before administering 200000 IU dose. The use of HD should be avoided if patients have sufficient vitamin D status at baseline. In such cases, lower doses of vitamin D would be effective and preferred. Our findings manifested that supplemented patients showed substantial improvements in the vitamin D status. The doses of 5000 IU/day and 200000 IU stat are comparable but the high dose necessitate monitoring. Moreover, the dose of 200000 IU stat carries an advantage of compliance and can effectively be used with vigorous monitoring of any toxicity.

Prevalence of VDD was evidently associated with various factors including practice of veil, limited sun-exposure and ethnicity in South Asian countries [36, 52–57]. Similar to the other studies [32, 53, 58], factors such as exposure to sunlight and use of sunblock are found to be independent predictor of vitamin D insufficiency the current study (Table 5). We analysis revealed that patients having average daily sun exposure of less than 15 minutes portend high propensity of vitamin D insufficiency. Similarly, the use of sunblock before going outside increase the risks of vitamin D insufficiency by four times. These findings underscore that in addition to supplementation, pregnant female must be encouraged to have adequate sun-exposure which could be beneficial for attaining the optimal serum 25(OH)D [59]. Findings of the previous investigations with significant association of sun-exposure with insufficient vitamin

D level were consistent with this study [32]. However, dressing habits and impact of vitamin D rich diet did not show any significant association with VDD in the present study. In contrast to the previous studies, there was no association between BMI and vitamin D status during our analysis. It is pertinent to mention that women in Pakistan spend most of their time indoor due to household activities and cultural norms. This indoor time further increases among pregnant women due to common beliefs of rest and restricted movement for baby care. These cultural or societal norms must be considered during the interpretation of results.

It is important to note that VDD has been found to be positively associated with low socio-economic status [60]. However, these findings are contrary to the results of the present study. The majority of the patients in our study were from good socioeconomic status but still had a high prevalence of VDD similar to another study conducted in Pakistan in which nursing mothers belonged to upper socioeconomic class [61]. These findings urge the need of education and awareness as a pivotal key to reduce the growing encumbrance of VDD in Pakistan. Patients should be educated about the significance of supplementation, factors associated with VDD and sources of vitamin D. Educational campaigns and patient counselling at the antenatal visit regarding VDD could be of paramount importance for pregnant women. A clinical pharmacist can play a crucial role in this regard.

Screening for VDD at gestation and implementation of vitamin D supplementation could be considered during antenatal care. Policy makers, nutritionists and other healthcare professionals can establish their roles to increase the awareness regarding VDD consequences. Moreover, food fortification of staple food should be initiated at National level. Health programs creating awareness regarding sun-exposure, dietary modification and supplementation should be initiated at both public and private healthcare facilities.

## Study limitations and strengths

This study was a single-centered study, results of which cannot be extrapolated to larger population. Current study was only conducted in Lahore city and hence requirements of women living in rural areas, other provinces, and different latitudes could be different. Inter-laboratory variation may have affected the serum 25(OH)D value of an individual as different cut-off reference values and varying techniques may have been used in different laboratories. No follow-up of neonates was done to determine the effects of adequate maternal 25(OH)D levels on neonatal health. Follow-up with neonate and measuring cord 25(OH)D levels would have further enlighten the significance of maternal supplementation. It must be noted that most of the study participants were recruited during the winter and autumn months of the year in which UV index is comparatively low and most of the women reside in their homes due to the cold waves. In this context, caution must be carried out to interpret the results for summer recruits. Last but not least, we tried to rigorously adjust the co-variates during the analysis. However, data on many important confounders which may affect vitamin D levels were missing for many patients due to observational nature of the study. This confounding affect can be explicitly adjusted in randomized controlled trials. Nonetheless, equal distribution of co-variates among treatment groups in the current study minimizes the risks of bias. We suggest careful consideration of this limitation during the interpretation of results and validation of findings. Moreover, there was no validated form used to assess the patient compliance with the regimen but compliance was reassured through self-reporting. Future studies incorporating these limitations are direly suggested.

Despite aforementioned shortcomings, this study provides data on prevalence of vitamin D specifically in Pakistani pregnant population and contains important information which can be used to address appropriate supplementation regimens in the country which could be

translated into improved vitamin D status during gestation. Findings of the current study may provide the basis to formulate guidelines and recommendations for vitamin D supplementation among pregnant women. Considering the dearth of regional investigations in Pakistan, results of the present study will serve to strengthen the field of research in the country. This study generates the hypothesis that vitamin D supplementation at a dose of 5000 IU/day during pregnancy is superior to the other regimens. However, well-controlled randomized trials are needed to confirm these findings.

## Conclusions

Current study suggests high proportion of VDD among pregnant women in Pakistan. Antenatal vitamin D supplementation proved to be an effective intervention and may benefit all the VD insufficient pregnant female. The stat dose of 200000 IU is equally effective as 5000 IU/day dose and also carries additional advantage of compliance but the propensity of drug toxicity cannot be disregarded. Future research should evaluate neonatal consequences of VDD and determine any association between the vitamin D status and BMI. There is a dire need to have randomized control trials (RCTs) to ascertain the effectiveness of various dosing regimens of vitamin D in mothers and neonates.

## Author Contributions

**Conceptualization:** Nida Bokharee, Yusra Habib Khan, Tauqeer Hussain Mallhi.

**Data curation:** Nida Bokharee, Abdulaziz Ibrahim Alzarea, Aisha Khokhar.

**Formal analysis:** Nida Bokharee, Tauqeer Hussain Mallhi, Nasser Hadal Alotaibi, Muhammad Shahid Iqbal, Kanwal Rehman, Abdulaziz Ibrahim Alzarea, Aisha Khokhar.

**Funding acquisition:** Nida Bokharee.

**Investigation:** Yusra Habib Khan, Tayyiba Wasim, Nasser Hadal Alotaibi.

**Methodology:** Yusra Habib Khan, Tauqeer Hussain Mallhi, Nasser Hadal Alotaibi, Muhammad Shahid Iqbal, Kanwal Rehman, Aisha Khokhar.

**Project administration:** Tayyiba Wasim.

**Resources:** Tayyiba Wasim, Abdulaziz Ibrahim Alzarea.

**Supervision:** Yusra Habib Khan, Tayyiba Wasim.

**Writing – original draft:** Nida Bokharee.

**Writing – review & editing:** Tauqeer Hussain Mallhi, Nasser Hadal Alotaibi, Muhammad Shahid Iqbal, Abdulaziz Ibrahim Alzarea, Aisha Khokhar.

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
