## [Decision Letter · Decision Letter 0]

13 Dec 2019

PONE-D-19-29625

Daily versus Stat Vitamin D Supplementation During Pregnancy; A prospective Cohort Study

PLOS ONE

Dear Dr Khan,

Thank you for submitting your manuscript to PLOS ONE. After careful consideration, we feel that it has merit but does not fully meet PLOS ONE’s publication criteria as it currently stands. Therefore, we invite you to submit a revised version of the manuscript that addresses the points raised during the review process.

We would appreciate receiving your revised manuscript by Jan 24 2020 11:59PM. To enhance the reproducibility of your results, we recommend that if applicable you deposit your laboratory protocols in protocols.io, where a protocol can be assigned its own identifier (DOI) such that it can be cited independently in the future. For instructions see: http://journals.plos.org/plosone/s/submission-guidelines#loc-laboratory-protocols

We look forward to receiving your revised manuscript.

Kind regards,

Frank T. Spradley

Academic Editor

PLOS ONE

2. Please provide a sample size and power calculation in the Methods, or discuss the reasons for not performing one before study initiation.

Reviewers' comments:

Reviewer's Responses to Questions

**Comments to the Author**

1. Is the manuscript technically sound, and do the data support the conclusions?

Reviewer #1: Yes

Reviewer #2: Yes

Reviewer #3: Yes

2. Has the statistical analysis been performed appropriately and rigorously? 

Reviewer #1: No

Reviewer #2: Yes

Reviewer #3: Yes

3. Have the authors made all data underlying the findings in their manuscript fully available?

Reviewer #1: Yes

Reviewer #2: Yes

Reviewer #3: Yes

4. Is the manuscript presented in an intelligible fashion and written in standard English?

Reviewer #1: No

Reviewer #2: Yes

Reviewer #3: Yes

5. Review Comments to the Author

Reviewer #1: This study, conducted in Pakistan, evaluate the effectiveness of three different regimes of vitamin D (VD) supplementations among pregnant women. Some details should be clarified before further evaluation.

1. Pregnant women were stratified into four groups (one control group, and three VD intervention groups). Were they randomly allocated to each group? If not, how was the dose of VD determined for each woman?

2. How long was the interval of side effects of VD supplementation monitored?

3. There were two methods employed to measure VD levels. The reason for using two different methods should be described. Besides, did the author assess the difference between the two methods. Moreover, the author said vitamin D3 (likely to be 25(OH)D3) were measured to assess the 25(OH)D concentrations. Why 25(OH)D2 was not measured.

4. Fisher exact test should be noted in the Tables if such a test was used. The Chi square test (or Fisher exact test) provided only one P-value, why there were multiple P-values provided for each category of a variable in Table 1 (also Table 4). Gravidity is defined as the number of times that a woman has been pregnant, which is equal to the “Total Number of pregnancies” in the table. Parity in Table 1 should better be presented as a categorical variable.

5. The difference in mean VD level at baseline and follow-up between all groups were tested using one-way ANOVA. Were the data satisfied the assumptions of one-way ANOVA?

6. The differences in VD levels between baseline and follow-up within each treatment group were tested, while the differences among groups were not tested. Thus, it is not rigorous to conclude that the group with 5,000 IU/d VD supplementation had the highest level of VD increment. Additionally, season is a crucial influencing factor of VD levels, and the authors should take it into account when analyzing and interpreting the result.

7. Multivariate analysis, other than Chi square test, should be performed to test the risk factors between women with sufficient and insufficient VD.

8. The sum of each category of the variable in Table 4 was not equal to the total number of each arm. The authors should recheck the data carefully.

9. English writing should be improved. There are numerous typos and grammar errors (a lot of missing articles).

Reviewer #2: First study to explore the impact of various vitamin D antenatal supplementation regimens among pregnant women in Pakistan.

1) Information about Food source of vitamin D should be added on data collection. Have the authors asked about the participants’ diet? Food frequency questionnaire could be applied.

2) Maternal vitamin D status should be followed through pregnancy. Comparison between early and late gestation might be important.

3) Follow up with neonates should be addressed and/or umbilical cord blood could be used to measure levels of 25(OH)D.

4) Explain how the maternal blood was collected and processed.

5) Deficiency of Vitamin D has been associated with risks for preeclampsia. Have the authors collected information about blood pressure?

6) The follow reference should be added on the current paper. PMID: 31669079; DOI: 10.1016/j.jand.2019.07.002

Reviewer #3: This study is important because it indicates the amount of vitamin D needed to achieve normal levels. Vit D could plan an important role to control inflammation and hypertension during pregnancy.

What does the G stand for “Patients received various regimens of vitamin D including 2000 IU/day (G1), 5000 IU/day (G2) and stat 200000 IU (G3). The g should define group I thought it was a typo

Can the authors retrieve data on blood pressure or any inflammatory markers. Could the prevalence of hypertension be determined. Could extrapolations to fetal weight or health be made?this would make the paper much more exciting.

The clinical methods description is very vague, there is now detail about how blood was collected.

6. PLOS authors have the option to publish the peer review history of their article (what does this mean?). If published, this will include your full peer review and any attached files.

Reviewer #1: Yes: Yunxian Yu, Bule Shao

Reviewer #2: No

Reviewer #3: No

---

## [Author Response · Author response to Decision Letter 0]

8 Jan 2020

Point-by-Point response to reviewers

Respected Editor,

We have received revisions/suggestions for our submitted manuscript. All the concerns and suggestions of the reviewers have been addressed and we hope that the revised version of the manuscript will satisfy the concerns of all reviewers. We have attached point-by-point response to each reviewer and revised version of manuscript is highlighted to track the changes. Please let us know if any other changes are required in this regard. Last but not least, we are very thankful to editor and reviewers for their time and efforts to put their valuable suggestions. Indeed their recommendations made this manuscript more scientifically elegant and sound. 

Reviewer 1

This study, conducted in Pakistan, evaluate the effectiveness of three different regimes of vitamin D (VD) supplementations among pregnant women. Some details should be clarified before further evaluation.

Query – 1: Pregnant women were stratified into four groups (one control group, and three VD intervention groups). Were they randomly allocated to each group? If not, how was the dose of VD determined for each woman?

Response – 1: Respected Reviewer, Thank you very much for pointing out the ambiguity in the manuscript. All the pregnant women attending OPD of the Hospital and fulfilling the inclusion criteria during the study period were consecutively included into the analysis. Patients were classified into four groups according to the VD dose they received during their visit. The choice of vitamin D dose was beyond our control and was at the complete discretion of the treating physician. Due to the observational nature of the study, it was not possible for us to randomly allocate the patients into the different treatment groups. This information on patients’ stratification has been elaborated and updated in the manuscript. We hope that the correct version would be satisfactory. There is another study (Wenisch et al.) comparing three different dosage regimens of antibiotic which has followed the similar methodology.

• Wenisch JM, Schmid D, Kuo H-W, Allerberger F, Michl V, Tesik P, et al. Prospective observational study comparing three different treatment regimes in patients with Clostridium difficile infection. Antimicrobial agents and chemotherapy. 2012;56(4):1974-8.

Query – 2: How long was the interval of side effects of VD supplementation monitored?

Response – 2: Respected Reviewer, side-effects of VD supplementation was monitored throughout the follow-up period of the study (2 months). This information has been updated in the manuscript as well.

Query – 3: There were two methods employed to measure VD levels. The reason for using two different methods should be described. Besides, did the author assess the difference between the two methods. Moreover, the author said vitamin D3 (likely to be 25(OH)D3) were measured to assess the 25(OH)D concentrations. Why 25(OH)D2 was not measured.

Response – 3: Respected Reviewer, Thank you very much for highlighting the ambiguity. There was only one method used to measure VD status, using serum total 25(OH)D level. Use of 1,25(OH)2D and 25(OH)D tests to assess the vitamin D status is still debatable. However, in order to address this concern, studies conducted by Kennel et al (2010) and Lips et al (2007) can be used as reference. In these studies, authors concluded that although 1,25(OH)2D is the active form of vitamin D, serum 25(OH)D is the barometer for the vitamin D status of an individual due to its longer half-life. Heaney (2004) also considered 25(OH)D to be the best indicator of Vitamin D status.

The technique used to ascertain the vitamin D status in the current manuscript was chemiluminescent immunoassay (CLIA), a quantitative method which measures total 25(OH)D and did not quantify D2 and D3 separately. This method is widely used by clinicians to assess Vitamin D status during their practice (Moon et al., 2016, Yu et al., 2009). CLIA has been used in various studies and a recent systematic review by Gallo et al., provides the utility of this method in wide range of studies. We hope that our response is satisfactory to address the concern of reviewer. Following is the list of references discussed in this response.

• Kennel KA, Drake MT, Hurley DL. Vitamin D deficiency in adults: when to test and how to treat. Mayo Clinic proceedings. 2010;85(8):752-8.

• Lips P. Relative value of 25 (OH) D and 1, 25 (OH) 2D measurements. Journal of Bone and mineral Research. 2007;22(11):1668-71.

• Moon RJ, Harvey NC, Cooper C, D'Angelo S, Crozier SR, Inskip HM, et al. Determinants of the Maternal 25-Hydroxyvitamin D Response to Vitamin D Supplementation During Pregnancy. The Journal of Clinical Endocrinology & Metabolism. 2016;101(12):5012-20.

• Yu CKH, Sykes L, Sethi M, Teoh TG, Robinson S. Vitamin D deficiency and supplementation during pregnancy. Clinical Endocrinology. 2009;70(5):685-90.

• Gallo S, McDermid JM, Al-Nimr RI, Hakeem R, Moreschi JM, Pari-Keener M, et al. Vitamin D Supplementation during Pregnancy: An Evidence Analysis Center Systematic Review and Meta-Analysis. J Acad Nutr Diet. 2019.

Query – 4: Fisher exact test should be noted in the Tables if such a test was used. The Chi square test (or Fisher exact test) provided only one P-value, why there were multiple P-values provided for each category of a variable in Table 1 (also Table 4). Gravidity is defined as the number of times that a woman has been pregnant, which is equal to the “Total Number of pregnancies” in the table. Parity in Table 1 should better be presented as a categorical variable.

Response – 4: Respected Reviewer, Thank you for correcting the mistake regarding the p-values. Based on your suggestion, all the p values are corrected and only one value is presented for more than 2 by 2 variables. Chi square test or Fisher exact test are described in the methodology and have been indicated in the tables, where applicable.

Query – 5: The difference in mean VD level at baseline and follow-up between all groups were tested using one-way ANOVA. Were the data satisfied the assumptions of one-way ANOVA?

Response – 5: Respected Reviewer, the groups were compared using One-Way ANOVA or Kruskal-Wallis Test depending on the data distribution. There were only four continuous variables (Mean VD level at baseline, Mean VD level at follow-up, age and BMI) and all of them were normally distributed.

Query – 6: The differences in VD levels between baseline and follow-up within each treatment group were tested, while the differences among groups were not tested. Thus, it is not rigorous to conclude that the group with 5,000 IU/d VD supplementation had the highest level of VD increment. Additionally, season is a crucial influencing factor of VD levels, and the authors should take it into account when analyzing and interpreting the result.

Response – 6: Respected Reviewer, the difference among the groups was also tested using paired t-test. Table 3 shows the difference at baseline and follow-up within each treatment group. Please refer to the Table 2 in the manuscript which shows the difference among the groups in the baseline as well as during follow-up. However, seasonal variation was not considered during the study, which should have been taken into account. However the data was collected in the relatively colder months of year in the country. Moreover, the data was collected in Peak gynae season in Pakistan which occurs from September to Feb 2019. To increase the applicability of the study and to get sufficient number of patients this study was conducted in these months. However, we have taken into account the potential impact of this season on VD. Since November to February is coldest period in the country in which sun exposure increases, we suggested considering such impact during the interpretation of results. Such impact of influencing factors has been discussed in the discussion section of the manuscript.

Query – 7: Multivariate analysis, other than Chi square test, should be performed to test the risk factors between women with sufficient and insufficient VD.

Response – 7: Respected Reviewer, Thank you very much for the recommendation. Based on your suggestion, we performed logistic regression analysis to ascertain the independent predictors associated with vitamin D insufficiency. The manuscript has been updated accordingly. Please refer to the Result where a new table (Table No. 5) is added. All the variables included for the analysis were either statistically tested or those having clinical plausibility with the outcome of interest.

Query – 8: The sum of each category of the variable in Table 4 was not equal to the total number of each arm. The authors should recheck the data carefully.

Response – 8: Respected Reviewer, Thank you very much for pointing out the mistakes in Table 4, the information has been updated in the manuscript.

Query – 9: English writing should be improved. There are numerous typos and grammar errors (a lot of missing articles).

Response – 9: Respected Reviewer, The manuscript has been proofread by native English speaker. Numerous typos, grammar and syntax errors have been addressed. We hope that the current version will satisfy the Reviewer`s concerns regarding the readability of the manuscript.

REVIEWER # 2

First study to explore the impact of various vitamin D antenatal supplementation regimens among pregnant women in Pakistan

Query - 1: Information about Food source of vitamin D should be added on data collection. Have the authors asked about the participants’ diet? Food frequency questionnaire could be applied.

Response – 1: Respected Reviewer, the data regarding the intake of vitamin D rich food source were collected as the part of data collection. Food frequency questionnaire was used to obtain the patient’s data on the food intake prior to the supplementation. However the data was not used previously. Relevant data has been used and the manuscript has been updated accordingly. As per you suggestion, this data is added in Table 1 of the manuscript.

Query – 2: Maternal vitamin D status should be followed through pregnancy. Comparison between early and late gestation might be important.

Response – 2: Respected reviewer, thank you so much for highlighting this important issue. Undoubtedly, data on pregnancy outcomes and gestational timing is of great importance. Unfortunately, we were unable to extract this information due to time constraints. Since this study was a postgraduate project to secure Master degree, investigators were only able to collect limited data required to test pre-defined objectives due to restricted time to complete the project. All the procedures or tasks related to the current project including synopsis writing, proposal defense, ethical approval, data collection, data analysis, drafting of results and submission of the report were supposed to be completed in one year of student candidature. In this context, only limited data was retrieved to achieve the predefined study objectives. However, to strengthen the impact of the data, vitamin D status was assessed twice (at baseline and during follow-up). Te objective of current studies were determined by keeping in view the current problem faced by Pakistan i.e. which dose of vitamin D is effective to maintain optimal vitamin D status in pregnant women. A similar study is conducted by Kaloczi et al (2014) where vitamin D levels were measured twice (at baseline and 28th week of gestation), as we did in our study. In another study by Moon et al (2016) assessed the vitamin D after 34th weeks of gestation. We were only able to include follow-up data of two months due to the limitations as described above. However, we have mentioned this limitation in the manuscript. We will definitely incorporate this suggestion in our future clinical trials we have planned in order to assess the effectiveness of vitamin D dosages in varying population of pregnant women. The trial will be initiated in this year. Following are the list of references discussed above in this response.

• Kaloczi LD, Deneris A. Rate of Low Vitamin D Levels in a Low-Risk Obstetric Population. Journal of Midwifery & Women's Health. 2014;59(4):405-10.

• Moon RJ, Harvey NC, Cooper C, D’Angelo S, Crozier SR, Inskip HM, et al. Determinants of the maternal 25-hydroxyvitamin D response to vitamin D supplementation during pregnancy. The Journal of Clinical Endocrinology & Metabolism. 2016;101(12):5012-20.

Query – 3: Follow up with neonates should be addressed and/or umbilical cord blood could be used to measure levels of 25(OH)D.

Response – 3: Respected reviewer, thank you so much for highlighting this important issue. Undoubtedly, data on neonatal outcomes and vitamin D status is of paramount importance. Unfortunately, we were unable to extract this information due to time constraints, as described in the previous query. However, we have addressed these limitations in the limitation section of the study. Furthermore, we would like to underscore that previous studies also did not record the cord blood 25(OH)D level as the study objectives were not extended to the measure the neonatal outcome of vitamin D supplementation. Kaloczi et al (2014) conducted a study in which 25(OH)D level was measured initially at first visit and at 28th week of gestation to evaluate effectiveness of vitamin D supplementation and no cord blood was tested. Moreover, the hospital setup was a non-charity structure institution and co-investigators could not bear the cost of the test as it was much expensive. However, these recommendations will be incorporated in future studies that we are planning. We are designing a clinical trial to be conducted in 2020 based on the current findings and we are in the phase of getting funding for this project. Your recommendations of measuring the levels of 25(OH)D throughout the gestation and at the time of delivery (using cord blood) will definitely be incorporated to improve the study and its impact in clinical practice. Following are the references used to support the query.

• Kaloczi LD, Deneris A. Rate of Low Vitamin D Levels in a Low-Risk Obstetric Population. Journal of Midwifery & Women's Health. 2014;59(4):405-10.

Query – 4: Explain how the maternal blood was collected and processed.

Response – 4: Respected Reviewer, Thank you for highlighting the missing information. Informed patient consent was obtained before the commencement of the study. Purpose of the study was briefed to the patients. 5 milliliters of maternal blood sample was obtained for the assessment purposes. Vitamin D status was assessed using Immunoassay technique i.e. Chemiluminescence Immunoassay Analyzer (CLIA). The detailed process of maternal blood collection has been added in the manuscript (under the heading methodology, sub-heading of Biochemical Analysis.

“Biochemical Analysis:

Vitamin D3 estimation was performed by chemiluminescence or CLIA (Chemiluminescence Immunoassay Analyzer) technique using state of the art Maglumi® 600 fully automated system. Maternal blood (5 milliliters) was collected at baseline and again at the follow-up. Blood samples were centrifuged and stored at -80° C followed by Chemiluminescence to assess the 25(OH)D level (ng/ml). Serum Vitamin D3 concentration was measured in Nano grams per milliliter (ng/ml). The cut-off reference points used to define vitamin D status in this study were < 20 ng/ml as deficiency, 20 to < 30 ng/ml as insufficiency, 30-100 ng/ml as sufficiency and > 100 ng/ml as toxicity (3-14). Maternal serum calcium levels were assessed using spectrometry method at the follow-up to rule out any manifestation of vitamin D intoxication.” 

Query – 5: Deficiency of Vitamin D has been associated with risks for preeclampsia. Have the authors collected information about blood pressure?

Response – 5: Respected Reviewer, Blood pressure was measured by attending physician in the clinic but was not recorded in the file. However, BP measurements were recorded for those diagnosed with pre-eclampsia or eclampsia. The diagnosis of pre-eclampsia was mentioned on the file. We re-checked the patients` record and were unable to find any case of pre-eclampsia for the present study.

Since the primary purpose of the current study was to compare the supplementation regimen, obstetric complications such as pre-eclampsia was not considered for the current study. Blood pressure was measured by the attending physician but not documented unless the patient was diagnosed with pre-eclampsia or eclampsia. The diagnosis was mentioned on the files of the patients. However in this study, no case of pre-eclampsia or eclampsia was present. We have again checked the patient record and found no such data with the indication of pre-eclampsia. However, we would like to underscore that various studies conducted earlier with a primary objective of supplementation comparison during gestation also did not record the Blood pressure as the parameter to be studied. Hollis et al (2011) conducted a study and primary objective was to evaluate safety and effectiveness of Vitamin D during pregnancy; however less attention was paid to the epidemiological features and other parameters associated with VDD. Similarly, other studies conducted by Rodda et al (2015), Cooper et al (2016) and Kaloczi et al (2014) did not record these parameters throughout the study as their primary objective was comparing of dosing regimen. Since your comments are of much value and helpful for us, any further suggestion/guidance in the analysis will be warmly welcomed. Following are the list of references provided to support the current query.

• Kaloczi LD, Deneris A. Rate of Low Vitamin D Levels in a Low-Risk Obstetric Population. Journal of Midwifery & Women's Health. 2014;59(4):405-10.

• Hollis BW, Johnson D, Hulsey TC, Ebeling M, Wagner CL. Vitamin D supplementation during pregnancy: double-blind, randomized clinical trial of safety and effectiveness. Journal of bone and mineral research : the official journal of the American Society for Bone and Mineral Research. 2011;26(10):2341-57.

• Rodda CP, Benson JE, Vincent AJ, Whitehead CL, Polykov A, Vollenhoven B. Maternal vitamin D supplementation during pregnancy prevents vitamin D deficiency in the newborn: an open-label randomized controlled trial. Clinical Endocrinology. 2015;83(3):363-8.

• Cooper C, Harvey NC, Bishop NJ, Kennedy S, Papageorghiou AT, Schoenmakers I, et al. Maternal gestational vitamin D supplementation and offspring bone health (MAVIDOS): a multicentre, double-blind, randomised placebo-controlled trial. The Lancet Diabetes & Endocrinology. 2016;4(5):393-402.

Query – 6: The follow reference should be added on the current paper. PMID: 31669079; DOI: 10.1016/j.jand.2019.07.002

Response – 6: Respected Review, Thank you very much for the recommendation. The recommended reference has been added in the discussion part of the manuscript.

REVIEWER # 3

Query-1: This study is important because it indicates the amount of vitamin D needed to achieve normal levels. Vit D could plan an important role to control inflammation and hypertension during pregnancy.

Response-1: Respected reviewer, thank you for acknowledging the significance of the current study. Though the impact of vitamin D on inflammation control and hypertension is of significant value, but we were unable to analyze such factors in the current study due to several reasons. First, data on blood pressure was not available on patient`s file and record. In our hospital, attending physician check blood pressure in the clinic and most of the time does not record the readings in the patient`s file. Such readings are recorded for patients with pre-eclampsia and there was no diagnosis of pre-eclampsia in the current study. Secondly, due to financial constraints inflammatory markers were not determined for the current study. Since the primary objective of the current study was to compare the dosing regimens of supplementation, the least attention was paid to other confounders. However, in align to your recommendations we will consider these parameters in our future study which we will start in 2020. We are planning to conduct a clinical trial based on the current findings and your recommendations will definitely be incorporated in the objectives of the study.

Query-2: What does the G stand for “Patients received various regimens of vitamin D including 2000 IU/day (G1), 5000 IU/day (G2) and stat 200000 IU (G3). The g should define group I thought it was a typo

Response-2: Respected reviewer, Thank you so much for pointing out the mistake. We have addressed this typo throughout the manuscript. The G represents the Group:

G = Group; G1 = Group 1 with 2000 IU dose, G2 = Group 2 with 5000 IU dose and G3 = Group 3 with 200,000 IU dose

Query-3: Can the authors retrieve data on blood pressure or any inflammatory markers. Could the prevalence of hypertension be determined. Could extrapolations to fetal weight or health be made?this would make the paper much more exciting.

Response-3: Respected reviewer, these variables were not considered before the commencement of the study. Blood pressure was measured by attending physician in the clinic but was not recorded in the file on every visit. However, BP measurements were recorded for those diagnosed with pre-eclampsia or eclampsia. The diagnosis of pre-eclampsia was mentioned on the file. We re-checked the patient record and were unable to find any case of pre-eclampsia for the present study. Moreover, as previously described, the study the study was purely observational in nature and ordering new tests was beyond the protocol. Moreover, the hospital where current study is conducted in no-charity institution and ordering of such tests was quite costly for patients as well as researchers.

Moreover, we would like to underscore that various studies conducted earlier with a primary objective of supplementation comparison during gestation also did not record the Blood pressure during the study period. Hollis et al (2011) conducted a study of similar nature and their primary objective was to evaluate safety and effectiveness of Vitamin D during pregnancy; however less attention was paid to the other obstetric complications and their association with VDD. Some other studies conducted by Rodda et al (2015), Cooper et al (2016) and Kaloczi et al (2011) also did not record these parameters throughout the study as their primary objective was regimen comparison. Moreover, the data regarding the inflammatory markers is not a part of routine tests in our hospital, so we were unable to retrieve the data. Last but not least, current project was part of postgraduate study for which we have only one year to plan a study, conducting a study, preparing results and submission of research report. Therefore, neonatal and gestational outcomes were not evaluated in the current study. However, we have incorporated your suggestions in our upcoming project which will start in 2020 as a doctorate degree project. Following is the list of references used to support the current query.

• Kaloczi LD, Deneris A. Rate of Low Vitamin D Levels in a Low-Risk Obstetric Population. Journal of Midwifery & Women's Health. 2014;59(4):405-10.

• Rodda CP, Benson JE, Vincent AJ, Whitehead CL, Polykov A, Vollenhoven B. Maternal vitamin D supplementation during pregnancy prevents vitamin D deficiency in the newborn: an open-label randomized controlled trial. Clinical Endocrinology. 2015;83(3):363-8.

• Cooper C, Harvey NC, Bishop NJ, Kennedy S, Papageorghiou AT, Schoenmakers I, et al. Maternal gestational vitamin D supplementation and offspring bone health (MAVIDOS): a multicentre, double-blind, randomised placebo-controlled trial. The Lancet Diabetes & Endocrinology. 2016;4(5):393-402.

Query-4: The clinical methods description is very vague, there is now detail about how blood was collected.

Response-4: Respected reviewer, the clinical methods description is revised in the manuscript. The detail of blood sample collection has been added to the manuscript; under sub-heading of “outcome measures”, in the Methodology section. We hope the current version will satisfy the concern of the Reviewers.

At the end, we are greatly thankful to all three reviewers for their valuable time and suggestions. Indeed their recommendations helped us a lot to make the current manuscript more scientifically sound and correct. Any further suggestions related to the current manuscript is warmly welcomed.

Dr Yusra Habib Khan

Corresponding author

---

## [Decision Letter · Decision Letter 1]

3 Feb 2020

PONE-D-19-29625R1

Daily versus Stat Vitamin D Supplementation During Pregnancy; A prospective Cohort Study

PLOS ONE

Dear Dr Khan,

Thank you for submitting your manuscript to PLOS ONE. After careful consideration, we feel that it has merit but does not fully meet PLOS ONE’s publication criteria as it currently stands. Therefore, we invite you to submit a revised version of the manuscript that addresses the points raised during the review process.

We would appreciate receiving your revised manuscript by Mar 19 2020 11:59PM. To enhance the reproducibility of your results, we recommend that if applicable you deposit your laboratory protocols in protocols.io, where a protocol can be assigned its own identifier (DOI) such that it can be cited independently in the future. For instructions see: http://journals.plos.org/plosone/s/submission-guidelines#loc-laboratory-protocols

We look forward to receiving your revised manuscript.

Kind regards,

Frank T. Spradley

Academic Editor

PLOS ONE

Reviewers' comments:

Reviewer's Responses to Questions

**Comments to the Author**

1. If the authors have adequately addressed your comments raised in a previous round of review and you feel that this manuscript is now acceptable for publication, you may indicate that here to bypass the “Comments to the Author” section, enter your conflict of interest statement in the “Confidential to Editor” section, and submit your "Accept" recommendation.

Reviewer #1: (No Response)

2. Is the manuscript technically sound, and do the data support the conclusions?

Reviewer #1: Yes

3. Has the statistical analysis been performed appropriately and rigorously? 

Reviewer #1: Yes

4. Have the authors made all data underlying the findings in their manuscript fully available?

Reviewer #1: Yes

5. Is the manuscript presented in an intelligible fashion and written in standard English?

Reviewer #1: Yes

6. Review Comments to the Author

Reviewer #1: 

The authors have addressed most of the questions. However, some were yet to be solved. 

Season or month of the year at the baseline should be taken into account throughout all analyses, given it is a crucial influencing factor of vitamin D (VD) level.As is shown in Table 2, VD levels at the baseline and follow-up were compared among the four groups, respectively. Besides, paired t-tests were performed to examine the VD increment after VD supplementation (Table 3) within each group. However, we would like to know whether the difference of VD increment among the four groups is statistically different (especially between G2 and G3). Thus, the VD increment of each group should be compared, and a multivariate model should be employed to control the confounding of baseline VD level, and more importantly, the season (or month of the year) at the baseline, as well as other potential confounders. Did the authors assess the compliance of each intervention group regarding VD supplementation?In the **Statistical Analysis section**, "Fischer exact test" is supposed to be “Fisher exact test.”

7. PLOS authors have the option to publish the peer review history of their article (what does this mean?). If published, this will include your full peer review and any attached files.

Reviewer #1: Yes: Yunxian Yu, Ph.D. Bule Shao

---

## [Author Response · Author response to Decision Letter 1]

9 Mar 2020

RESPONSE TO REVIEWERS

Respected Editor,

 We have received second round of revisions/suggestions for our submitted manuscript. All the concerns and suggestions of the reviewer have been addressed and we hope that the revised version of the manuscript will satisfy the concerns of all reviewer. We have attached point-by-point response to the reviewer and revised version of manuscript is also highlighted to track the changes. Please let us know if any other changes are required in this regard. Last but not least, we are very thankful to editor and reviewers for their time and efforts to put their valuable suggestions. Indeed their recommendations made this manuscript more scientifically elegant and sound. 

REVIEWER # 1

Query – 1: Season or month of the year at the baseline should be taken into account throughout all analyses, given it is a crucial influencing factor of vitamin D (VD) level. 

Response – 1: Respected reviewer, thank you for the recommendation. The study was carried out in the colder months of the country i.e. October 2018 to April 2019 [Autumn = 22nd September – 21st December, Winter = 22nd December – 20th March, Spring = 21st March - 21st June]. A total of 112 patients were recruited in Autumn season, 104 in Winter and 65 in Spring. The UV index (UVI) is comparatively low in these months. The optimal UVI for the production of Vitamin D (D-UV) is greater than 3.1,2 However, the UVI decreases from October to December and then raises from January to April. Moreover, Chi-square test was preformed to check association of season with the VD level at the baseline; the test was insignificant (p = 0.072). With respect to your conerns, we have also highlighted the same issue in the limitation section of the manuscript.

1. Holick MF. Vitamin D Deficiency. N Engl J Med 2007; 357: 266-81.

2. Hollick MF, Chen TC. Vitamin D deficiency: a worldwide problem with health consequences. Am J Clin Nutr 2008; 87: 1080S-6S.

Query – 2: As is shown in Table 2, VD levels at the baseline and follow-up were compared among the four groups, respectively. Besides, paired t-tests were performed to examine the VD increment after VD supplementation (Table 3) within each group. However, we would like to know whether the difference of VD increment among the four groups is statistically different (especially between G2 and G3). Thus, the VD increment of each group should be compared, and a multivariate model should be employed to control the confounding of baseline VD level, and more importantly, the season (or month of the year) at the baseline, as well as other potential confounders. 

Response – 2: Respected Reviewer, Thank you for the suggestion. VD increment between all the groups has been checked as per your recommendation. All groups shows that there is a significant difference between the control group and treatment cohorts (G1, G2 and G3); (p < 0.001). The treatment groups were also compared with each other and there was statistically significant difference [G1 and G2; (p < 0.001); G1 and G3; (p < 0.001)]. However, there was no statistically significant difference between G2 and G3 (p = 0.579). This information has been updated in the manuscript in the text before Table 3. To the best of our knowledge, if covariates are equally distributed between the treatment groups then confounder adjustments are not necessary during the analysis. However, we respect your suggestion and carried out regression analysis in which we adjusted the season to determine the potential contributors of vitamin D insufficiency among study participants.

Query – 3: Did the authors assess the compliance of each intervention group regarding VD supplementation?

Response – 3: Respected Reviewer, patient compliance was self-reported by patients and was assessed during the follow-up visit. All the patients in the treatment groups were compliant with their medication regimen.

Query – 4: In the Statistical Analysis section, “Fischer exact test” is supposed to be “Fisher exact test.”

Response – 4: Respected Reviewer, Thank you for pointing out the typo. Manuscript has been updated accordingly.

Corresponding author: Yusra Habib Khan

---

## [Decision Letter · Decision Letter 2]

13 Mar 2020

PONE-D-19-29625R2

Daily versus Stat Vitamin D Supplementation During Pregnancy; A prospective Cohort Study

PLOS ONE

Dear Dr Khan,

Thank you for submitting your manuscript to PLOS ONE. After careful consideration, we feel that it has merit but does not fully meet PLOS ONE’s publication criteria as it currently stands. Therefore, we invite you to submit a revised version of the manuscript that addresses the points raised during the review process.

SPECIFIC ACADEMIC EDITOR COMMENT: There are few minor comments that still need addressing by the authors. 

We would appreciate receiving your revised manuscript by Apr 27 2020 11:59PM. To enhance the reproducibility of your results, we recommend that if applicable you deposit your laboratory protocols in protocols.io, where a protocol can be assigned its own identifier (DOI) such that it can be cited independently in the future. For instructions see: http://journals.plos.org/plosone/s/submission-guidelines#loc-laboratory-protocols

We look forward to receiving your revised manuscript.

Kind regards,

Frank T. Spradley

Academic Editor

PLOS ONE

Reviewers' comments:

Reviewer's Responses to Questions

**Comments to the Author**

1. If the authors have adequately addressed your comments raised in a previous round of review and you feel that this manuscript is now acceptable for publication, you may indicate that here to bypass the “Comments to the Author” section, enter your conflict of interest statement in the “Confidential to Editor” section, and submit your "Accept" recommendation.

Reviewer #1: (No Response)

2. Is the manuscript technically sound, and do the data support the conclusions?

Reviewer #1: Partly

3. Has the statistical analysis been performed appropriately and rigorously? 

Reviewer #1: No

4. Have the authors made all data underlying the findings in their manuscript fully available?

Reviewer #1: Yes

5. Is the manuscript presented in an intelligible fashion and written in standard English?

Reviewer #1: Yes

6. Review Comments to the Author

Reviewer #1: 1. In the abstract, please provide the VDD frequency after intervention.

2. Conclusion has better be simplified.

3. While the effect of VD supplement was compared between groups, the VD influence factors such as sun-exposure time, Use of sunblock and baseline VD level must be adjusted. This part is the main result for evaluating the effect of intervention. Then add this part to manuscript.

7. PLOS authors have the option to publish the peer review history of their article (what does this mean?). If published, this will include your full peer review and any attached files.

Reviewer #1: No

---

## [Author Response · Author response to Decision Letter 2]

25 Mar 2020

RESPONSE TO REVIEWERS

Respected Editor,

 We have received third round of minor revisions/suggestions for our submitted manuscript. All the concerns and suggestions of the reviewer have been addressed and we hope that the revised version of the manuscript will satisfy the concerns of the reviewer. We have attached point-by-point response to the reviewer and revised version of manuscript is also highlighted to track the changes. Please let us know if any other changes are required in this regard. Last but not least, we are very thankful to editor and reviewers for their time and efforts to put their valuable suggestions. Indeed their recommendations made this manuscript more scientifically elegant and sound. 

REVIEWER # 1

Query – 1: 1. In the abstract, please provide the VDD frequency after intervention.

Response – 1: Respected Reviewer, VDD frequency has been added in the abstract section.

Query – 2: 2. Conclusion has better be simplified.

Response – 2: Respected Reviewer, as per your suggestion we have simplied the conclusion section of the manuscript. I hope that current version of conclusion will satisfy the concern.

Query – 3: While the effect of VD supplement was compared between groups, the VD influence factors such as sun-exposure time, Use of sunblock and baseline VD level must be adjusted. This part is the main result for evaluating the effect of intervention. Then add this part to manuscript.

Response – 3: Respected Reviewer, the confounding factors has been adjusted. This resulted in addition of another table controlling the impact of covariates on outcome variable. We have also modified the results according to the new table. We really hope that the current version of manuscript will satisfy your concern in this regard.

At the end, are are greatly thankful for your time and interest in our research . Indeed, your suggestions were quite valuable for us. Our believe on peer review is more strenghthened amid healthy discussion pertaining to current research.

Corresponding author: Yusra Habib Khan

---

## [Editor Report · Decision Letter 3]

27 Mar 2020

Daily versus Stat Vitamin D Supplementation During Pregnancy; A prospective Cohort Study

PONE-D-19-29625R3

Dear Dr. Khan,

We are pleased to inform you that your manuscript has been judged scientifically suitable for publication and will be formally accepted for publication once it complies with all outstanding technical requirements.

With kind regards,

Frank T. Spradley

Academic Editor

PLOS ONE

---

## [Editor Report · Acceptance letter]

3 Apr 2020

PONE-D-19-29625R3 

Daily versus Stat Vitamin D Supplementation During Pregnancy; A prospective Cohort Study 

Dear Dr. Khan:

I am pleased to inform you that your manuscript has been deemed suitable for publication in PLOS ONE. Congratulations! Your manuscript is now with our production department. 

With kind regards,

on behalf of

Dr. Frank T. Spradley 

Academic Editor

PLOS ONE